# Mammals show faster recovery from capture and tagging in human-disturbed landscapes

Wildlife tagging provides critical insights into animal movement ecology, physiology, and behavior amid global ecosystem changes. However, the stress induced by capture, handling, and tagging can impact post-release locomotion and activity and, consequently, the interpretation of study results. Here, we analyze post-tagging effects on 1585 individuals of 42 terrestrial mammal species using collar-collected GPS and accelerometer data. Species-specific displacements and overall dynamic body acceleration, as a proxy for activity, were assessed over 20 days post-release to quantify disturbance intensity, recovery duration, and speed. Differences were evaluated, considering species-specific traits and the human footprint of the study region. Over 70% of the analyzed species exhibited significant behavioral changes following collaring events. Herbivores traveled farther with variable activity reactions, while omnivores and carnivores were initially less active and mobile. Recovery duration proved brief, with alterations diminishing within 4–7 tracking days for most species. Herbivores, particularly males, showed quicker displacement recovery (4 days) but slower activity recovery (7 days). Individuals in high human footprint areas displayed faster recovery, indicating adaptation to human disturbance. Our findings emphasize the necessity of extending tracking periods beyond 1 week and particular caution in remote study areas or herbivore-focused research, specifically in smaller mammals.

Wildlife movement studies are essential for understanding animal behavioral responses to global environmental changes, sustaining ecosystem functioning, and successful nature conservation[1]. Animal movements are pivotal in shaping biodiversity patterns and connecting habitats[2–4]. Comprehending the far-reaching anthropogenic influence on movements is therefore paramount to effective land use planning and conservation strategies[5]. In wildlife research, GPS telemetry is increasingly applied to study animal movement ecology[6,7]. Current devices track individual movements at unprecedented levels of spatial and temporal precision. Beyond high-resolution motion tracking, modern technology allows for a variety of sensors to be attached to animals, pushing animal tracking into the realm of Big Data[7–9] and providing researchers with information not only of animals' whereabouts but also offering insights into for example their activity patterns, derived from accelerometers[10]. Such sensors measure static and dynamic acceleration, representing animal movement in three dimensions[11] and allowing for the quantification of activity or proxies thereof[12–15]. One well-established proxy is the overall dynamic body acceleration (ODBA), which estimates activity-related energy expenditure of free-ranging animals[16,17].

While tracking devices have enabled researchers to collect invaluable data on animal movements and behavior[18] and have yielded considerable scientific and conservation benefits[5,19], concerns have been raised about potential adverse effects of these devices on study animals[20–23]. Deploying telemetry sensors on animals involves capturing, handling, and releasing the focal individual[24,25]. The effects of physical capture, chemical immobilization, and restraint of animals on possible post-release behavioral modifications are, however, understudied in wildlife species, which may affect the welfare of animals and the interpretation of study results (but see: refs. 26–28). The capture

✉e-mail: stiegler@uni-potsdam.de

and handling process involves several stress-inducing and physically demanding events that are attributable to human presence and may involve sudden or loud noises, social isolation, limited movement, and impaired vision[29–31]. The use of neurologically active chemicals can affect animal behavior and movement for several days[25,32], ultimately triggering behavioral changes[33–35]. These behavioral changes can negatively affect home range formation and activity patterns[26], body condition[32,36,37], and even reproductive success and survival[38,39].

In long-term deployments of GPS tracking devices over several months or years, omitting data from the initial days post-capture is common to reduce the chances of biased results driven by capture effects. However, this assumes that animals will have adapted to the attached sensor after this period. Yet, a lack of data exists on the response and recovery of animals to capture and how it varies among species. In contrast, for short-term deployments of several days (e.g., bats or flying foxes), the effects of stress from the collaring process on animal behavior and activity, in addition to the physical impairment effects due to the tag weight[40], may result in biased findings with animals having insufficient time to recover. Considerable strides have been made in reducing stress during capture and improving the weight and comfort of devices[35]. However, effectively evaluating and minimizing adverse collaring impacts remain complex tasks that demand increased research attention[41]. A few studies have examined the effects of collar deployment procedures and tags on animal behavior (see, e.g., refs. 42,43), but general ethical guidelines for acceptable practices regarding attached devices remain unresolved[44]. Furthermore, there is a notable absence of protocols for handling data during the initial tracking days due to uncertainty surrounding the duration over which animal-borne tracking devices impair individuals. Compounding this challenge is the difficulty in determining and evaluating 'normal' behavior[22], with only a limited number of case studies attempting this. In captive scimitar-horned oryx *Oryx dammah*, Stabach et al.[43] showed elevated stress hormone levels for up to 5 days and behavioral changes (e.g., increased headshaking) for up to 3 days in collared individuals. Van de Bunte[45] found that collared red pandas *Ailurus fulgens* in captivity reduced daily activity levels and food intake compared to non-collared individuals. In free-ranging red deer *Cervus elaphus*, Becciolini et al.[46] found increased movement rates and avoidance of their center of activity for up to 10 days, likely reflecting recovery from effects of the deployment procedure. In Eurasian beavers *Castor fiber*, the body mass of dominant individuals decreased considerably with repeated capture events[47]. American black bears *Ursus americanus* tended to avoid human presence after capture events[48]. Similarly, roe deer *Capreolus capreolus* reduced activity and were displaced towards woodland to avoid human disturbance. These behavioral changes decreased during the first 10 days, with females being less sensitive than males[26].

The effects of capture (e.g., helicopter darting or capture, chasing, trapping) on animal behavior can vary widely across species, sex, tag size, and type, deployment duration, the specific deployment procedure, or the environment[22,26,44,49]. Species differ in stress responses, especially throughout the initial days of tracking, and the time taken to return to their normal behavior[31,37]. Notably, animals living in anthropogenic landscapes adapt their space use and become more tolerant to human disturbances[50,51]. As such, the effects of deployment procedures likely differ between individuals who are behaviorally adapted to different levels of human proximity. In a meta-analysis, Samia et al.[52] found populations habituated to human stressors to be more tolerant towards human disturbance. Consequently, since movement[53] and behavior[54,55] change with human proximity, we expect altered responses to capture and immobilization. In this study, we aggregated high-resolution GPS tracking and acceleration (ACC) data from 42 terrestrial mammal species over time from capture to quantify the magnitude and duration of collaring impacts on movement activity. We developed three measures to quantify disturbance effects based on individual movement characteristics and activity (Fig. 1). First, we quantified disturbance intensity to assess changes in daily displacement and activity (measured as ACC/ODBA) by calculating the sum of deviations for each of the initial ten tracking days from the subsequent 10-day average. Second, we calculated recovery speed, reflecting an individual's adaptability during the first 10 days, using the slope of the disturbance intensity curve on day one. Third, we calculated recovery duration using the time when each individual returned to their long-term average for each behavioral metric. Even though anesthesia dosage is calculated per kilogram, we assume that larger mammals experience a more pronounced disturbance, as they often require longer durations of anesthesia[24] and face more substantial physiological challenges during immobilization such as hyperthermia[56] compared to smaller mammals. This prolonged exposure can lead to more significant physiological and behavioral disruptions. In contrast, owing to their higher energy requirements relative to their body size, smaller mammals need to be consistently more active.

Capturing and tagging animals acts as a manipulative experiment, enabling us to investigate how animals respond to such disturbances. Beyond data exclusion considerations, such studies allow for the exploration of various hypotheses related to patterns in the behavioral responses of animals. We hypothesize that responses to capture are not only species-specific but may also encompass broader patterns driven by distinct traits. Due to their reproductive roles, we expect females to exhibit heightened sensitivity and more gradual recovery. We also assumed that dietary requirements are reflected in the responses, as herbivores may be more flexible in finding forage, allowing them to find shelter to recover. In contrast, carnivores need to roam continuously for survival and are evolutionary less adapted to being hunted, potentially making them more susceptible to prolonged impairment. By comprehensively examining this aspect, we expect the diet to influence the duration and intensity of an individual's impairment. Furthermore, we were particularly interested in whether animals in remote areas with fewer anthropogenic influences recover slower because individuals are less adapted to human disturbances.

In this work, we quantified the overall disturbance impact of capture across terrestrial mammal species, including assessing the time required for recovery and identifying periods most affected, which could bias the interpretation of results if not adequately accounted for. More than 70% of the species analyzed showed behavioral changes following collaring events. Herbivores traveled larger distances, while omnivores and carnivores were less active and mobile during the initial days post-release. Recovery duration proved brief, with alterations diminishing within 4–7 tracking days for most species, with individuals in high human footprint areas displaying faster recovery, indicating adaptation to human disturbance.

## Results
### Disturbance intensity
Among the 42 terrestrial mammal species analyzed, 30 were sensitive to the collaring procedure and significantly changed their activity or displacement behavior during the first 10 days after release (Fig. 2, Table 1, Figs. S1–S30). In total, we found that 25 of 41 species increased or decreased their activity, and 19 out of 40 species changed their displacements during the first 10 days of tracking (Fig. 2, Table 1). While within-species variability was high ($p_{ID_{GPS}}$ and $p_{ID_{ACC}}$ <0.001), sex did not significantly influence species-specific reaction behavior ($p_{sex_{GPS}}$ and $p_{sex_{ACC}}$ >0.05). On the first day of tracking, individuals were, on average, less active compared to their long-term mean ($-7.8 \pm 19.2$%; mean ± SD), whereas daily displacements were higher ($6.9 \pm 23.8$%; mean ± SD), with large SD attributed to strong intra- and interspecific variability. Net deviations, i.e., absolute deviations on the first day, were $14.2 \pm 15$% for activity and $18.7 \pm 16.3$% for displacements. The activity level of

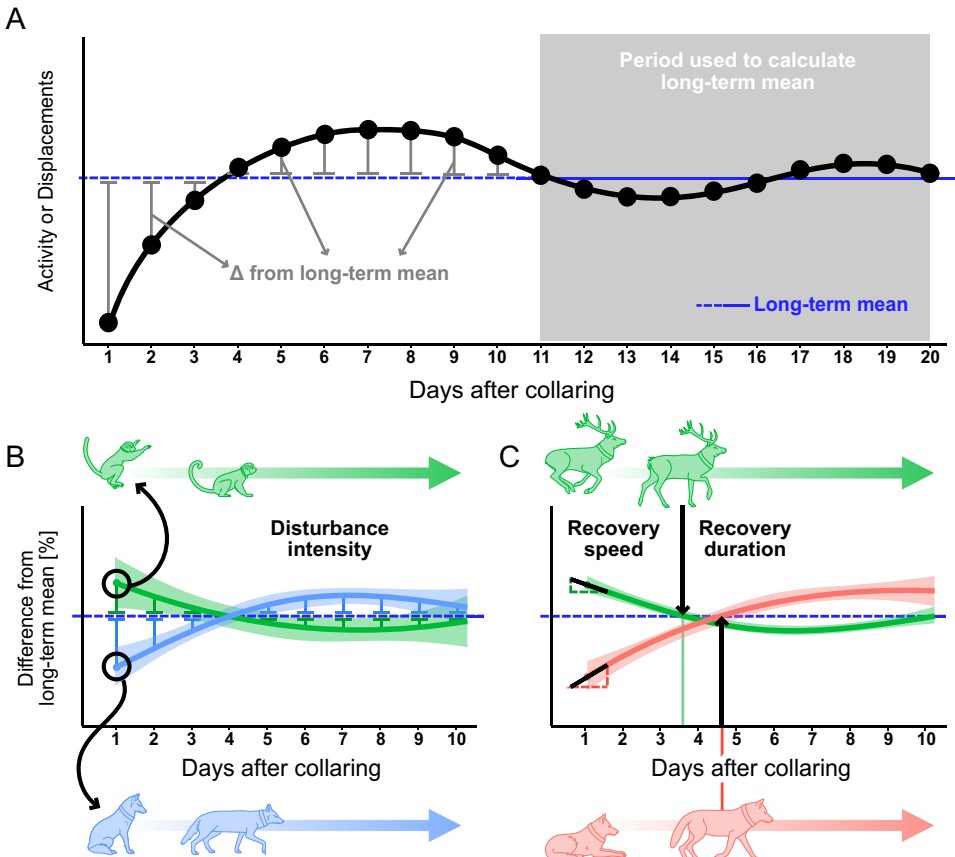

**Fig. 1 | Methods calculating the disturbance intensity, recovery speed and duration with specific examples. A** Illustrates the difference of daily activity (ODBA) and displacements (days 1–10) from the long-term means (days 11–20). First, we calculated daily (days 1–10) activity (ODBA) and displacements. Subsequently, we related derived values to the long-term mean (days 11–20). The analysis was conducted identically for activity and displacements. **B** To calculate the disturbance intensity, we related daily averaged values (displacement, activity) to the respective mean during days 11–20. The upper example illustrates the disturbance intensity of *Propithecus verreauxi*, with increased displacements on the first days, before converging towards the long-term mean; the lower illustrates the disturbance intensity in activity of *Canis aureus*, with decreased activity during the initial days of tracking. **C** Recovery speed was calculated as the |slope| on day one post-release, and recovery duration was determined as the time when animals reverted to their long-term mean for the first time post-release. The upper example illustrates the recovery speed and duration in activity of *Cervus elaphus*, the lower one of *Canis lupus*.

25 species differed substantially immediately after release compared to subsequent days, with a gradual stabilization during the initial days (Table 2). This trend was particularly evident in omnivores ($R^2 = 0.374$, Dev. explained = 46.4%). While omnivores and carnivores were less active during the initial days, pooled herbivore data revealed both increased and decreased activity rates. A similar pattern was found for displacements, as most species traveled longer distances after collaring events compared to the long-term mean (days 11–20; $R^2 = 0.25$, Dev. explained = 37.7).

On the first day post-release, moose (*Alces alces*) exhibited the largest increases in displacement distance, moving 63% further compared to the long-term mean, followed by common eland *Tragelaphus oryx* (52%), and spotted hyena *Crocuta crocuta* (44%). In contrast, leopards *Panthera pardus* were found to have the largest reductions in displacement distances, reducing their movement distances by-65%, followed by wolves *Canis lupus* (−44%), and Eurasian lynx *Lynx lynx* (−43%). Moose also had the largest increases in activity on day one (44%), followed by red deer *Cervus elaphus* (26%), and Mongolian khulan *Equus hemionus hemionus* (9%). Wolves had the largest decreases in activity on day one (−48%), followed by the white-tailed mongoose *Ichneumia albicauda* (−41%), and leopard *Panthera pardus* and golden jackal *Canis aureus* (−41%). In general, carnivores traveled shorter distances post-release, aside from the spotted hyena (*Deviance day*1$_{GPS}$ = 44%) and fossa *Cryptoprocta ferox* (*Deviance day*1$_{GPS}$ = 12%). In this study, we did not

investigate mortality rates as only individuals that survived for at least 20 days post-tagging were included in our analysis.

## Recovery speed and duration

Recovery speed in activity is best explained by a high human footprint index of the respective study site, the individuals' sex [+male] (Fig. 3AB, Table 3) and a larger body mass (competing model, $\Delta AIC < 2$, Tab. S2). A fast recovery in displacements was best explained by the species-specific diet [+carnivore] and its body mass, with large species recovering considerably faster (Fig. 3CD, Table 4). During the first 10 days of tracking, the difference from the long-term mean of displacements decreased from 33 ± 17% on day 1 to 21 ± 20% on day 10, while activity decreased from 24 ± 14%–12 ± 6%; *deviance*$_{day1}$ *vs. deviance*$_{day10}$ for all species with $p \leq 0.05$, Fig. 3, Table 1. Comparing individual days in days 11–20 to the mean of this period indicated mean routine variations of 14% for activity and 35% for displacements.

Recovery duration also differed between dietary types. Omnivores and carnivores returned to their mean long-term behavior in both disturbance intensity measures after 5–6 days (Table 2), with data beyond this period being less influenced by collaring events. In contrast, herbivores were the quickest to return to their mean long-term displacement behavior but were slowest to return to their long-term activity levels: 3.6 ± 1.0 days (displacements), and 6.6 ± 0.9 days (activity), mean ± SD.

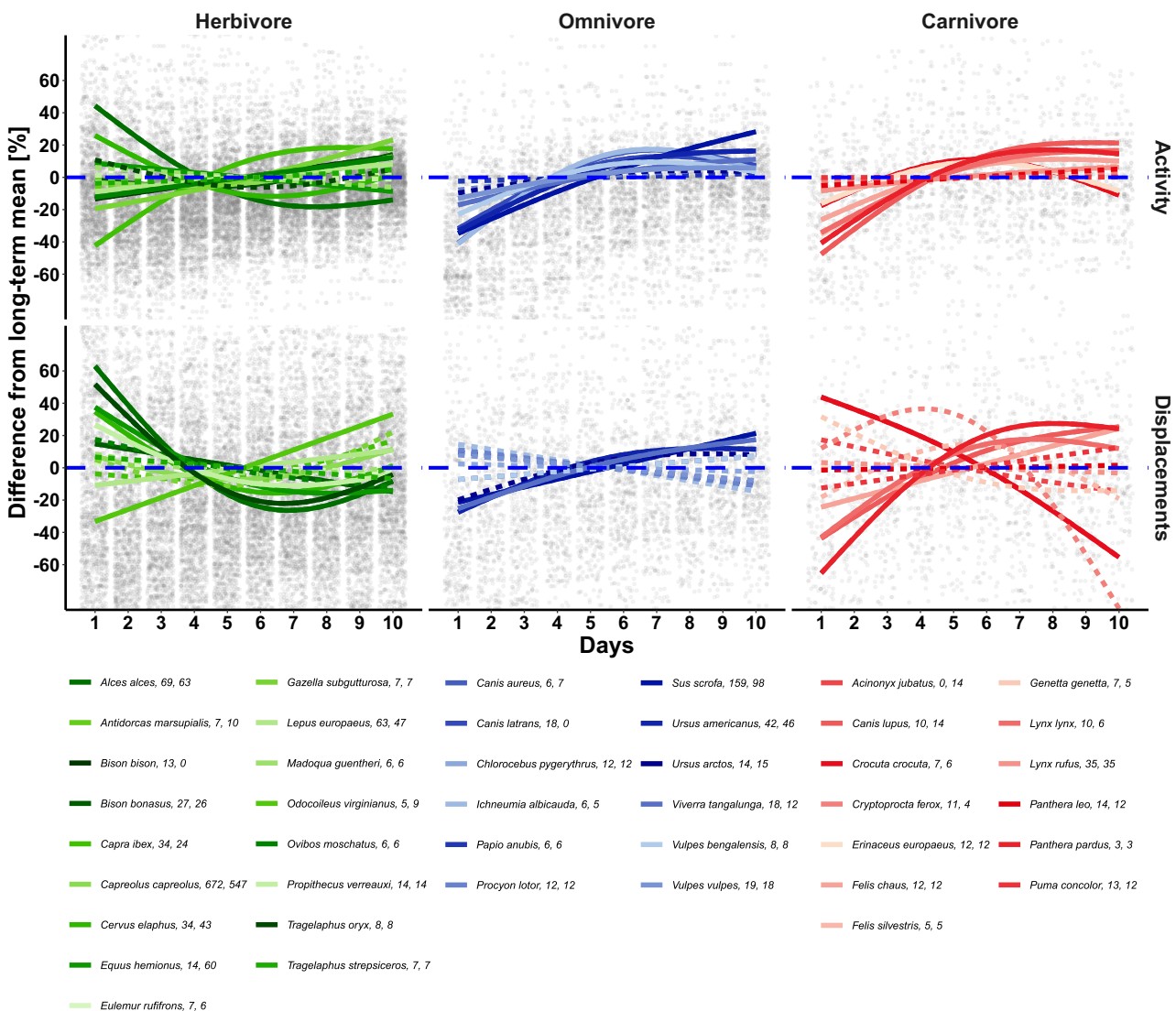

**Fig. 2 | Disturbance intensity: Impacts of collaring on activity and displacements during the initial 10 days post-release.** Daily differences to the long-term mean of activity (upper) and displacements (lower) split by diet: herbivores (left), omnivores (middle), and carnivores (right) for 42 mammal species, $n = 1585$. All species with $p \leq 0.05$ are shown as solid lines and species with $p > 0.05$ or $n < 5$ as dotted lines. Activity: $R^2 = 0.374$, Dev. explained = 46.4%, displacements: $R^2 = 0.25$, Dev. explained = 37.6%. Predictions are derived from two Generalized Additive Mixed Models with Gamma error distributions to assess the effect of disturbance intensity on activity and displacements of the focal species over time. The dotted blue line represents the long-term mean (average for days 11–20). In the legend following each species name, the first number refers to the number of individuals for activity and the second for displacements.

## Discussion

Our findings revealed widespread evidence of post-collaring behavioral changes in animal activity and displacements. Animals displayed a general trend in their responses, marked by the most pronounced deviations in behavior immediately following successful collar deployment. Subsequently, their behavior stabilized, converging on their long-term mean within four to 7 days (Tab. 2). This recovery duration represents the initial period of more pronounced data bias. Responses found in our dataset are consistent with the findings of case studies from the respective species: In moose *A. alces*, the observed reaction is in accordance with Neumann et al.[57], who identified larger spatial displacements for up to 4.5 days after capture. In wild boars *Sus scrofa*, similar to our findings, the first post-capture days were characterized by low activity and low mobility levels, which then gradually restored to stable levels at approximately 10 days[28]. Additionally, we observed increased movement rates for red deer immediately after release, as found by Becciolini et al.[46]. We can not make any conclusions about the effect of tagging on survival rates, as

only data from individuals that survived the study period were considered.

Males recovered on average 1.3 days faster than females from collaring-induced changes in activity, aligning with findings in roe deer[26], yet this effect was not detected in displacements. Females may require a longer recovery time due to gestation, birth, and rearing of offspring (as only 5–10% of mammalian species engage in paternal care[58]). These factors may aggravate negative impacts associated with the attachment of tracking devices, potentially leading to increased stress levels, reduced foraging efficiency, or, as a consequence, compromised reproductive success[26–28]. We expect this effect to be even more pronounced in pregnant or lactating females. However, due to the heterogeneous nature of the dataset, with various species captured over different times across continents, we did not account for an individual's physiological or behavioral season.

Omnivores and carnivores were generally less active than herbivores after release. In cases where animals are caught with bait, as is sometimes done for carnivores and omnivores, individuals may not

**Table 1 | Species model summary: Disturbance intensity in activity and displacements**

| Study species | | Activity | | | | Displacements | | | |
|---|---|---|---|---|---|---|---|---|---|
| Scientific Name | Common Name | edf | ref.df | Statistic | *p*-value | edf | ref.df | Statistic | *p*-value |
| *Acinonyx jubatus* | Cheetah | | | | | 1.160 | 1.294 | 3.799 | 0.058 |
| *Alces alces* | Moose | 1.993 | 2.000 | 190.955 | <0.001 | 1.994 | 2.000 | 76.011 | <0.001 |
| *Antidorcas marsupialis* | Springbok | 1.000 | 1.000 | 1.333 | 0.248 | 1.000 | 1.000 | 0.176 | 0.675 |
| *Bison bison* | American Bison | 1.000 | 1.000 | 11.711 | 0.001 | | | | |
| *Bison bonasus* | European Bison | 1.859 | 1.980 | 6.700 | 0.001 | 1.000 | 1.001 | 6.856 | 0.009 |
| *Canis aureus* | Golden Jackal | 1.904 | 1.991 | 14.369 | <0.001 | 1.000 | 1.000 | 0.441 | 0.507 |
| *Canis latrans* | Coyote | 1.940 | 1.996 | 25.793 | <0.001 | | | | |
| *Canis lupus* | Gray Wolf | 1.894 | 1.989 | 37.056 | <0.001 | 1.684 | 1.900 | 12.894 | <0.001 |
| *Capra ibex* | Alpine Ibex | 1.936 | 1.996 | 92.773 | <0.001 | 1.664 | 1.887 | 1.924 | 0.098 |
| *Capreolus capreolus* | Roe Deer | 1.996 | 2.000 | 150.162 | <0.001 | 1.996 | 2.000 | 30.008 | <0.001 |
| *Cervus elaphus* | Red Deer | 1.942 | 1.997 | 31.220 | <0.001 | 1.961 | 1.999 | 16.210 | <0.001 |
| *Chlorocebus pygerythrus* | Vervet Monkey | 1.000 | 1.000 | 0.302 | 0.583 | 1.000 | 1.000 | 0.082 | 0.775 |
| *Crocuta crocuta* | Spotted Hyena | 1.875 | 1.984 | 4.426 | 0.014 | 1.387 | 1.625 | 9.713 | <0.001 |
| *Cryptoprocta ferox* | Fossa | 1.000 | 1.000 | 0.335 | 0.563 | 1.886 | 1.987 | 8.911 | <0.001 |
| *Equus hemionus* | Mongolian Khulan | 1.000 | 1.000 | 5.465 | 0.019 | 1.925 | 1.994 | 30.232 | <0.001 |
| *Erinaceus europaeus* | European Hedgehog | 1.891 | 1.988 | 5.816 | 0.004 | 1.000 | 1.000 | 0.001 | 0.983 |
| *Eulemur rufifrons* | Red-fronted Lemur | 1.000 | 1.000 | 0.061 | 0.805 | 1.000 | 1.000 | 0.059 | 0.809 |
| *Felis chaus* | Jungle Cat | 1.812 | 1.965 | 14.060 | <0.001 | 1.000 | 1.000 | 8.527 | 0.004 |
| *Felis silvestris* | Wildcat | 1.000 | 1.000 | 1.337 | 0.248 | 1.754 | 1.939 | 1.323 | 0.262 |
| *Gazella subgutturosa* | Goitered Gazelle | 1.280 | 1.481 | 9.762 | <0.001 | 1.669 | 1.891 | 1.114 | 0.256 |
| *Genetta genetta* | Common Genet | 1.809 | 1.964 | 15.326 | <0.001 | 1.448 | 1.695 | 2.954 | 0.130 |
| *Ichneumia albicauda* | White-tailed Mongoose | 1.919 | 1.993 | 14.329 | <0.001 | 1.000 | 1.000 | 1.277 | 0.259 |
| *Lepus europaeus* | European Hare | 1.195 | 1.352 | 18.290 | <0.001 | 1.000 | 1.000 | 6.623 | 0.010 |
| *Lynx lynx* | Eurasian Lynx | 1.811 | 1.964 | 20.903 | <0.001 | 1.694 | 1.907 | 4.823 | 0.025 |
| *Lynx rufus* | Bobcat | 1.313 | 1.528 | 7.565 | 0.006 | 1.000 | 1.000 | 0.439 | 0.508 |
| *Madoqua guentheri* | Günther's dik-dik | 1.000 | 1.000 | 1.165 | 0.281 | 1.289 | 1.495 | 0.347 | 0.766 |
| *Odocoileus virginianus* | White-tailed Deer | 1.000 | 1.000 | <0.001 | 0.993 | 1.000 | 1.000 | 18.475 | <0.001 |
| *Ovibos moschatus* | Muskox | 1.000 | 1.000 | 4.636 | 0.031 | 1.205 | 1.369 | 1.557 | 0.283 |
| *Panthera leo* | African Lion | 1.000 | 1.000 | 1.889 | 0.169 | 1.010 | 1.020 | 0.028 | 0.886 |
| *Panthera pardus* | Leopard | 1.758 | 1.941 | 8.783 | 0.001 | 1.685 | 1.901 | 5.692 | 0.013 |
| *Papio anubis* | Olive Baboon | 1.000 | 1.000 | 0.178 | 0.673 | 1.000 | 1.000 | 0.666 | 0.415 |
| *Procyon lotor* | Raccoon | 1.266 | 1.461 | 2.380 | 0.168 | 1.000 | 1.000 | 1.802 | 0.180 |
| *Propithecus verreauxi* | Verreaux's Sifaka | 1.000 | 1.000 | 0.099 | 0.753 | 1.718 | 1.921 | 4.117 | 0.041 |
| *Puma concolor* | Cougar | 1.000 | 1.000 | 0.001 | 0.971 | 1.000 | 1.000 | 2.255 | 0.133 |
| *Sus scrofa* | Wild Boar | 1.867 | 1.982 | 418.661 | <0.001 | 1.000 | 1.000 | 53.265 | <0.001 |
| *Tragelaphus oryx* | Gemsbok | 1.760 | 1.943 | 1.990 | 0.166 | 1.861 | 1.981 | 7.511 | 0.001 |
| *Tragelaphus strepsiceros* | Greater Kudu | 1.096 | 1.183 | 0.438 | 0.483 | 1.000 | 1.000 | 0.373 | 0.541 |
| *Ursus americanus* | American Black Bear | 1.928 | 1.995 | 76.150 | <0.001 | 1.826 | 1.970 | 13.861 | <0.001 |
| *Ursus arctos* | Brown Bear | 1.552 | 1.799 | 2.919 | 0.121 | 1.529 | 1.778 | 3.161 | 0.104 |
| *Viverra tangalunga* | African Civet | 1.563 | 1.809 | 12.381 | <0.001 | 1.369 | 1.602 | 5.142 | 0.026 |
| *Vulpes bengalensis* | Bengal Fox | 1.811 | 1.964 | 6.690 | 0.004 | 1.000 | 1.000 | 0.501 | 0.479 |
| *Vulpes vulpes* | Red Fox | 1.650 | 1.878 | 6.532 | 0.008 | 1.000 | 1.000 | 1.367 | 0.242 |
| s(ID) | | 1248 | 1451 | 6.584 | <0.001 | 1063 | 1261 | 4.589 | <0.001 |
| s(sex) | | 0.140 | 1.000 | 1.159 | 0.312 | 0.001 | 1.000 | <0.001 | 0.472 |
| R-sq. (adj) | | 0.374 | | | | 0.250 | | | |
| Deviance explained | | 46.4% | | | | 37.7% | | | |
| *n* | | 1452 | | | | 1262 | | | |

The presented results include values for estimated degrees of freedom (edf) and reference degrees of freedom (ref.df), where edf represents the effective degrees of freedom resulting from the fitted model, indicating model flexibility, while ref.df serves as a baseline measure for comparison. (see methods, Eq. (1)).

need to carry out foraging movements in the following days as they would under normal circumstances. A more proximate explanation for the reduced movement and activity could also be a reaction to chemical immobilization. In contrast, 65% of the herbivores increased their activity on the first day post-release. Resting to conserve energy does not seem like a legitimate reaction to being chased and immobilized because their natural response to being chased by predators is escaping by moving. The recovery speed of activity and displacements

after collaring events was slower in herbivores than omnivores and carnivores. From an evolutionary perspective, this is surprising since predators frequently chase many wild herbivores, and therefore, herbivores may be expected to be better adapted to and recover faster from disturbances. Yet, these responses may be offset by the potent anesthesia used, particularly for large herbivores (e.g., *Bison sp.*, *A. alces*, *Tragelaphus strepsiceros*, *C. elaphus*). For all species, we found

strong intraspecific variation in the response behavior, which may be context-specific or linked to animal personalities[59], traditionally assessed along a bold-shy continuum[60,61].

Stress-related activity of wildlife is often categorized as either fight or flight[62]. This can also hold true for the post-capture response of wildlife to either the capture event or the collar. Characterization of fight-flight was first identified in human psychology[63], but as Bracha et al.[64] noted, the addition of "freeze" to the term is needed. In wildlife, this can be extended to include hiding in response to disturbance[65]. Post-release behavior likely includes a complex blend of all these responses, as well as additional stressors they encounter during that timeframe. To add to this complexity, in places with significant anthropogenic influence, animals frequently display enhanced tolerance and adaptation to human presence[50,51]. Animals that adapt to human presence may experience reduced competition for resources compared to natural habitats[66].

Management practices, such as supplementary feeding, which can cause habituation and changes in space use, mobility, or activity (e.g.,

**Table 2 | Duration until return to mean long-term behavior (mean values ± standard deviation)**

| Dietary type | Mean days (ACC) | Mean days (GPS) |
|---|---|---|
| Herbivore | 6.59 ± 0.86 | 3.60 ± 1.00 |
| Omnivore | 5.50 ± 0.63 | 5.63 ± 0.49 |
| Carnivore | 5.09 ± 0.88 | 5.44 ± 0.50 |

Displacements were calculated based on localizations (GPS) and activity based on accelerometer data (ACC).

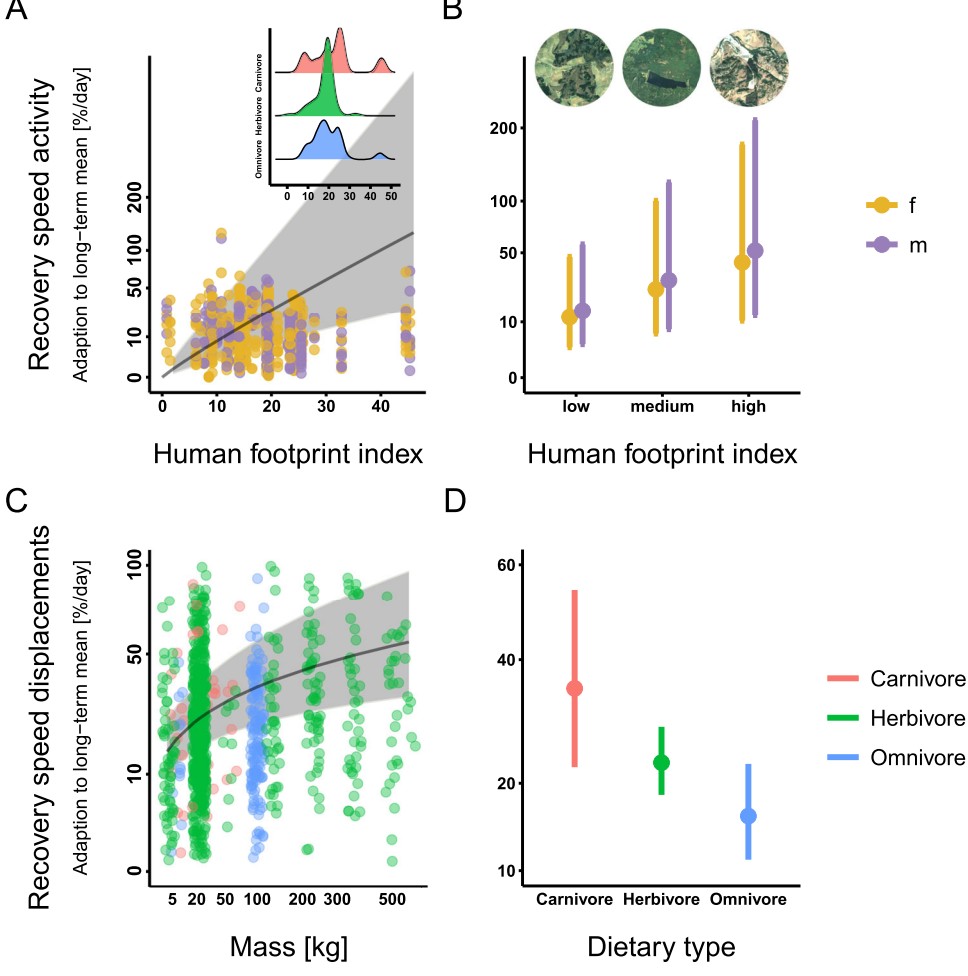

**Fig. 3 | Recovery speed described in relation to dietary type, an individual's sex, and the human Footprint index of the study area. A**, **B** Recovery speed (of activity) described in relation to sex and the Human Footprint index (HFi), *n* = 1241. High recovery speed values indicate a fast recovery. High HFi values indicate a strong anthropogenic influence, and low values indicate a high degree of remoteness. The inset (**A**) shows the density plots of the sample size distribution for each dietary guild in regard to HFi. **B** Predictions are presented for values of the lower (12.37), median (18.68), and upper (25) quartiles of HFi. Insets here (**B**) present exemplary satellite imagery of sites with differing HFi; left to right: an area with little infrastructure and some habitat fragmentation [HFi: 10]; agricultural fields with small forest patches, road infrastructure, and some settlements [HFi: 17]; a more

degraded landscape with a quarry and an adjacent solar park [HFi: 25] (©Landsat / Copernicus, GoogleEarth 2020-2023[88]). Landscapes with extreme HFi values (close to zero: representing pristine, undisturbed areas; close to 50: representing dense populated urban areas) were less present in the dataset and, as such, examples are not shown. **C**, **D** Recovery speed (of displacements) described in relation to body mass (**C**) and dietary type (**D**), *n* = 1014. Recovery speed describes the speed of change in activity or displacements as a percentage of the respective long-term mean on day one. Dots (**A**, **C**) represent calculated values. Dots (**B**, **D**) and the solid lines (**A**, **C**) represent mean modeled values, and bars (**B**, **D**) as well as the gray shaded area (**A**, **C**) are 95% confidence intervals. Note that the y-axis is sqrt-transformed.

**Table 3 | Recovery speed of activity**

| Recovery speed activity | | | |
|---|---|---|---|
| **Predictors** | **Estimates** | **CI** | **p** |
| (Intercept) | −5.51 | −10.08 – −0.94 | 0.018 |
| mass | 0.31 | −0.13 – 0.75 | 0.172 |
| sex [m] | 0.19 | 0.02 – 0.36 | 0.025 |
| diet [herbivore] | 0.60 | −1.25 – 2.45 | 0.527 |
| diet [omnivore] | −0.34 | −1.96 – 1.29 | 0.685 |
| HFi | 1.83 | 1.40 – 2.25 | <0.001 |
| **Random Effects** | | | |
| $\sigma^2$ | 2.02 | | |
| $\tau_{00\ study\ species}$ | 2.53 | | |
| ICC | 0.56 | | |
| $N_{study\ species}$ | 25 | | |
| **Observations** | 1241 | | |
| **Marginal** $R^2$ / **Conditional** $R^2$ | 0.111 / 0.605 | | |

The best-fit model to describe recovery speed in terms of activity spent included the species' body mass, sex, dietary type, and the study site's Human Footprint Index (HFi) as independent variables. Study species was implemented as a random effect (see methods, Eq. (2)).

**Table 4 | Recovery speed of displacements**

| Recovery speed displacements | | | |
|---|---|---|---|
| **Predictors** | **Estimates** | **CI** | **p** |
| (Intercept) | 0.70 | −0.47 – 1.86 | 0.240 |
| diet [herbivore] | −0.42 | −0.91 – 0.07 | 0.092 |
| diet [omnivore] | −0.78 | −1.35 – −0.23 | 0.006 |
| mass | 0.25 | 0.14 – 0.36 | <0.001 |
| **Random Effects** | | | |
| $\sigma^2$ | 1.31 | | |
| $\tau_{00\ study\ species}$ | 0.06 | | |
| ICC | 0.05 | | |
| $N_{study\ species}$ | 17 | | |
| **Observations** | 1014 | | |
| **Marginal** $R^2$ / **Conditional** $R^2$ | 0.065 / 0.107 | | |

The best-fit model to describe recovery speed in terms of displacements included dietary type and the species' body mass as independent variables. Study species was implemented as a random effect (see methods, Eq. (2)).

ref. 67), may also influence behavior after collaring. Furthermore, some species demonstrate behavioral flexibility and can adjust their activity patterns or habitat preferences[68] and their movement behavior[69] to avoid direct conflicts with humans. For example, some mammals, such as raccoons *Procyon lotor* and coyotes *Canis latrans*, thrive in urban areas by utilizing human-associated food resources and adapting their behavior to coexist with humans[50,51], yet, the impact of anthropogenic influence is species-specific[70]. Previous studies have shown that human interactions can strongly influence animal behavior. For example, the coexistence of humans and wildlife in urban areas often selects individuals with bold personalities[71–73]. On the other hand, animals inhabiting remote areas have less exposure to human presence and, consequently, encounters. Hence, when such animals encounter humans, they might show an exacerbated response toward the disturbance and remain alert for a prolonged time. While this assertion is speculative, it is supported by our finding here, where individuals in remote areas recovered slower from collaring than those in highly anthropogenically influenced areas. With numerous deployment methods like helicopter darting, chasing, or trapping being applied in the field, analyzing their effect was not feasible within the

scope of this study. The effect of the deployment method remains unclear, and the selected method may even change along an HFI gradient. For example, helicopter darting may be the only option in areas with little infrastructure, whereas in more urban areas, alternative options are preferred. Interpreting the effect of the human footprint should take into account that deployment type could be influenced by the respective study area. Therefore, we strongly recommend documenting these methodological decisions for future research.

There exists a fine balance between obtaining valuable data and ensuring the well-being of tracked animals. Researchers must consider these ethical dilemmas carefully and implement tracking methods that minimize harm and maximize animal welfare. Omitting initial data can contribute to reducing biased results, thereby generating more accurate outcomes that could better inform conservation efforts. Yet, it may be difficult to detect the effects of collars during short-term deployments, as the data obtained is highly time-constrained. While our study was confined to assessing behavioral alterations associated with collaring events, it is important to note that even short-term modifications in behavior can incur energetic costs, reduce energy intake, or influence predation risk and, as such, potentially impact animal survival and fitness[74–76]. As we only considered data from individuals that survived for at least 20 days post-tagging, we could not account for possible mortality rates. The inclusion of such data in future studies could contribute to an even more holistic understanding of the consequences of tagging.

While established animal welfare guidelines and regulatory requirements that allow for such invasive studies exist, many of these rely on findings from isolated case studies. Our study of post-release telemetry data of 42 terrestrial mammalian species reveals potential biases in wildlife GPS and ACC data during the initial days of animal tracking, likely due to invasive immobilization and tagging procedures, which may influence movement ecology findings. These impacts, however, fade within a relatively short time frame of four to 7 days, suggesting that the overall impact of collaring is minimal and short-lived, which is good news for animal tracking science. In studies where longer tracking is not feasible, researchers should be aware of these disturbance biases. Particularly, short-term studies, lasting <7 days, may be significantly compromised. These studies are prevalent in certain research areas, for example, where battery weight strongly limits tracking duration. Based on our findings, we strongly advocate extending animal tracking periods well beyond 7 days whenever possible. Further efforts relating the findings of this study to other important variables such as method of capture, type of tag, drug combinations, and post-release behavior could provide valuable insights into best practices in reducing capture myopathy, stress, and data bias. By understanding and addressing these limitations, researchers can maximize GPS-collaring advantages while limiting adverse effects on study animals. Undoubtedly, animal tracking will continue to contribute to our understanding of the environment, with progress in this field being propelled by ongoing technological developments, improved techniques, and heightened ethical considerations.

## Methods
### Data collection and preparation
Animal tracking data (GPS and ACC, see Supplementary Note 1 for permits) from multiple data providers were either directly sourced from tables or downloaded from the Movebank data repository[77] with the help of the R package Move[78]. In the first step, we omitted individuals with missing data during the initial 20 days, resulting in 1585 unique individuals. We defined data as missing if any discontinuation resulted in <1 GPS fix per hour and less than one activity measurement per 30 min. The resulting number of individuals per terrestrial mammal species ranged from 4 to 672 (mean $n_{acc} = 36.4$, mean $n_{gps} = 32.6$) out of total individuals $n_{acc} = 1452$ of 41 species across 57 study sites, and total individuals $n_{gps} = 1262$ of 40 species across 55 study sites.

We classified the data into two periods: the initial 10 days following the individual's release and days 11–20. We considered the latter timeframe representative of 'long-term' behavior, expecting that the response to the collaring/handling process had subsided within the initial 10 days, as shown in previous studies (e.g., *A. alces* ≤ 4.5 days[57], *C. capreolus* ≤ 10 days[26], *C. elaphus* ≤ 10 days[46]).

We calculated mean daily ODBA values for each individual with the R package moveACC[79] as ODBA = $|A_x| + |A_y| + |A_z|$ for tri-axial measurements; and as ODBA = $|A_x| + |A_y|$ for bi-axial measurements, where $A_x$, $A_y$, and $A_z$ are the derived dynamic accelerations corresponding to the three perpendicular axes of the sensor[13]. Downsampling from three to two axes to compare ACC measurements was not necessary, as the raw data were used per individual to calculate the disturbance intensity, which is then expressed in percent. Acceleration records obtained from individuals with only one axis (*Acinonyx jubatus*) were not considered. The temporal resolution of both GPS and ACC data was adjusted by rounding timestamps to the nearest 5 min interval. Then, displacements were calculated using the R package adehabitatLT[80] as each individual's mean displacement (m) from one GPS fix to the next within each 24 h interval. For each study site, we extracted the Human Footprint index (HFi:[81,82]) and calculated the mean HFi for a 5 km radius around the center of the study site (mean longitude, mean latitude).

### Disturbance intensity
Subsequently, we related daily averaged values (displacement, activity) to the respective mean during days 11–20 to calculate the disturbance intensity (Fig. 1). We applied two Generalized Additive Mixed Models with Gamma error distributions for the disturbance intensity in activity and displacements to estimate the effect on the focal species in combination with time (i.e., days 1–10) on daily differences to the long-term mean using the R package mgcv[83]. Since we did not expect a linear relationship, we specified the predictor variable time as a smooth term for each species and a first-order auto-regressive correlation structure corAR1 among the residuals of the model associated with each individual. Sex was included as a random smoothing effect, allowing for a smooth relationship between sex and the dependent variable. This allows for individual-specific effects of sex on the response, which can be useful when assuming that the relationship between sex and the response is not strictly linear but varies smoothly across individuals or species. The disturbance intensity model was specified as follows:

$$
\begin{aligned}
\text{deviance}_{id,t} &\sim \text{Gamma}(\eta_{id,t}, \alpha) \\
\eta_{id,t} &= \exp\left(f(t)_{\text{species}} + u_{id} + u_{sex} + \nu_{id,t}\right) \\
\nu_{id,t} &= \rho \nu_{id,t-1} + \epsilon_{id,t-1} \\
\epsilon_{id,t-1} &\sim \mathcal{N}(0, \sigma^2)
\end{aligned} \tag{1}
$$

Thus, the linear predictor $\eta_{id,t}$ includes an autoregressive process of order one (AR[1]). Here, the parameter $\rho$ accounts for the temporal autocorrelation, *id* represents the animal identifier, and *t* is the corresponding time point. In addition, *u* indicates the use of random intercepts. Deviance was calculated and modeled separately for both activity and displacement.

### Recovery speed and duration
For all individuals of species with significant disturbance effects (Fig. 2, Table 1), we calculated the |slope| on day one after the release as a measure of recovery speed, i.e., how fast individuals adapt throughout the first days. The slope was calculated for each individual as the first derivative for $x = 1$ from the ID-specific fitted curve with $y \sim log(x)$. Recovery speed, expressed in units of percentage per day, quantifies the rate of adaptation of individuals. The steeper the slope (i.e., the higher the values), the faster individuals were at adapting or acclimating. We applied separate linear mixed effect models for activity

and displacement to estimate the recovery speed in both activity and displacements, using the R package lme4[84] using the respective measurements, |slope day 1| as the dependent variable. We included sex, dietary type (herbivore, omnivore, carnivore), body mass derived from literature values[85] (Table S1), and the Human Footprint index of the study area as independent variables and study species as a random effect. Due to incomplete data and many different levels, we did not consider the deployment procedure as an independent variable. The dependent variable, as well as the independent variables, body mass and HFi, were log-transformed. The model was calculated using Gaussian error distribution and a natural logarithm link function. Subsequently, we selected models using the R package MuMIn[86]. By ranking model combinations via the Akaike Information Criterion (AIC), we considered all independent variables in the best-fit models within 2 AIC units in the final model and report the respective summary. Models were calculated using all gap-less data available for the independent and dependent variables, resulting in minor variations in sample size and species analyzed for activity and displacements.

To assess the stabilization period of collaring effects on activity and displacement, we used the fitted disturbance intensity model (Eq. 1) to calculate the period until individuals reverted to their average long-term behavior for both disturbance intensity measures (activity and displacements) for the first time post-release. For this, we included all individuals of species in which significant patterns were identified with the disturbance intensity model above.

The *recovery speed* model was specified as a linear mixed effect model:

$$
\begin{aligned}
\log(|\text{slope day1}|)_{id,\text{species},\text{studysite}} &\sim \mathcal{N}\left(\eta_{id,\text{species},\text{study site}}, \sigma^2\right) \\
\eta = \beta_0 + \beta_1 \log(\text{mass}_{\text{species}}) &+ \beta_2 \log(\text{HFi}_{\text{species},\text{study site}}) \\
+ \beta_{sex} \text{sex}_{id} &+ \beta_{diet} \text{diet}_{\text{species}} + u(\text{species}),
\end{aligned} \tag{2}
$$

where *sex* and *diet* were specified as categorical variables; *slopeday*1 was calculated and modeled separately for activity and displacement.

### Reporting summary
Further information on research design is available in the Nature Portfolio Reporting Summary linked to this article.

### Data availability
The datasets generated in this study to create the respective figures have been provided in the Source Data file. The GPS and acceleration datasets used and analyzed in this study are available in the Movebank Data Repository[77] at www.movebank.org (*Antidorcas marsupialis*, ID: 904829042; *Chlorocebus pygerythrus*, ID: 17629305; *Erinaceus europaeus*, ID: 354843286; ID: 348067475; ID: 490547558; ID: 1371906275; *Felis silvestris*, ID: 40386102; *Genetta genetta*, ID: 19814565; *Ichneumia albicauda*, ID: 158898881; *Lepus europaeus*, ID: 918554628; ID: 1138520346; ID: 4048590; ID: 25727477; ID: 43360515; ID: 71038468; ID: 73514179; *Lynx rufus*, ID: 501787846; ID: 475878514; *Panthera pardus*, ID: 17629305; *Papio anubis*, ID: 17629305; *Procyon lotor*, ID: 4048590; *Taurotragus oryx*, ID: 904829042; *Tragelaphus strepsiceros*, ID: 904829042; *Viverra tangalunga*, ID: 57540673; *Vulpes vulpes*, ID: 4048590; ID: 326682415; ID: 173932849); data from Euromammals[87] can be accessed by logging into their website or via a contact form at https://euromammals.org/ (*Capra ibex*; *Capreolus capreolus*; *Cervus elaphus*; *Lynx lynx*; *Sus scrofa*); or can be obtained from data providers upon request through the corresponding author. Source data are provided with this paper.

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

## Acknowledgements

We are very thankful for the support of Karatina University, Smithsonian Institution's National Museum of Natural History, and Mpala Research Centre; the staff of Polish and Slovak Tatra National Parks for their help in bear trapping; all Carpathian Brown Bear Project members who assisted in the field during captures, handling and data collection; Zbigniew Krasinski from the Białowieza National Park and Tomasz Kaminski from the Mammal Research Institute PAS for help in European bison collaring; the BioMove RTG including associated helpers in the field, the workers at the ZALF research station; the Forest and Wildlife Research Center at Mississippi State University; the members of Euromammals including Eurodeer, Euroboar, and Euroreddeer; Junta de Castilla y Leon, Gobierno Principado Asturias, Ministerio Transicion Ecologica, Tragsatec; the Oyu Tolgoi's Core Biodiversity Monitoring Program, implemented by the

WCS through a cooperative agreement with Sustainability East Asia LLC for their help in Khulan capture, marking and radiotracking; N. Sharma, G. Basson, D. Medeiros, J. McGraw, R. Reed, A. Johnston, H. Maschmeyer, R. King, B. Nichols, J. Suraci, for essential support in animal tracking; D. Simpson, S. Ekwanga, M. Mutinda, G. Omondi, W. Longor, M. Iwata, A. Surmat, M. Snider, W. Fox, and K. VanderWaal for field assistance, M. Crofoot, D. Rubenstein, and L. Frank for sharing their field equipment, and M. Kinnaird and T. Young for logistical support; L. Purchart, M. Kutal, J. Krojerová, and K. Purchartová for scientific background and project coordination and P. Forejtek for veterinarian support; the Danau Girang Field Centre research group in collaboration with the Sabah Wildlife Department, Veterinarian support provided by Drs. M. Gonzalez, S. Guerrero-Sanchez, D. Ramirez, L. Benedict, and P. Nagalingam; field assistants and personnel at Zackenberg Research Station; the Office Français de la Biodiversité, especially Jean-Luc Hamann and Vivien Siat and the Office National des Forets, including the wildlife technicians, the foresters, and the many volunteers for their help in the capture of red and roe deer; field collaborators and veterinarians of the Leibniz-IZW, Berlin, especially Janina Radwainski; the Namibian Ministry of Environment, Forestry and Tourism, the Namibian farmers, and the entire team of the Cheetah Research Project of the Leibniz-IZW, Berlin; K. Boyer, S. Peper, C. Wilson, Z. Johnson, H. Greenburg, K. Haydett, D. Warren, D. Payne, J. Hoffman, M. Proctor, J. Gaskamp for assistance with trapping wild pigs and white-tailed deer in Oklahoma; the University of California, Santa Cruz and the California Department of Fish and Wildlife for their partnership in the Santa Cruz Puma project; and all non-mentioned technicians, and workers in the field. This work was supported by the DFG-funded research training group "BioMove" (DFG-GRK 2118/1); by a National Science Foundation Postdoctoral Research Fellowship in Biology (DBI-1402456) awarded to Adam W. Ferguson and Paul W. Webala; the Polish-Norwegian Research program administered by the National Research Centre for Research and Development in Poland (POL-NOR/198352/85/2013), Tatra National Park own funding; by the German Federal Ministry of Education and Research BMBF within the Collaborative Project "Bridging in Biodiversity Science-BIBS" (grant number: 01LC1501); by the Polish Ministry of Sciences and Information Technology (grant no 2P04F 011 26); Frankfurt Zoological Society – Help for Threatened Wildlife and the EU LIFE program (project no LIFE06 NAT/PL/000105); by the DFG: KA 1082/17-1; by the DFG: KA 1082/16-1; by Safari Club International Foundation, Michigan Department of Natural Resources, and the Federal Aid in Wildlife Restoration Act under Pittman-Robertson project W-147-R; by grant QK1910462 and CZ021010.00.0160190000803; by Ministerio de la Transicion Ecologica; by the Peninsula Open Space Trust, Land Trust of Santa Cruz County, California Department of Fish and Wildlife, Santa Clara Open Space Authority; by the National Geographic Society Committee for Research and Exploration #9385-13; by the Washington University in Saint Louis ICARES grant 2015; by the Dean's office of the Faculty of Forestry and Wood Technology, Mendel University in Brno and Training Forest Enterprise Masaryk Forest Kr^tiny; by Houston Zoo; the Sime Darby Foundation; Ocean Park Conservation Foundation Hong Kong (TM01.1718); and Phoenix Zoo; by the 'Mov-It' Agence Nationale de la Recherche grant ANR-16-CE02-0010-02 to NM; by the Federal Ministry of Education and Research, Germany, FKZ: 01LL1804A; by the Office Français de la Biodiversité (OFB); by the "Stiftung Naturschutz Berlin"; by the Noble Research Institute, LLC; by 15. Juni Fonden and Copenhagen Zoo; by the Italian Ministry of Education, University and Research (PRIN 2010-2011, 20108 TZKHC, 418 J81J12000790001); by the Foreste Casentinesi National Park; by the Regione Autonoma della Sardegna, Provincia di Sassari, and Fondazione Banco di Sardegna; by the National Science Foundation; by the Messerli Foundation Switzerland; CS was supported by the Elsa-Neumann foundation; by the US National Science Foundation (grant nos. BCS 99-03949, BCS 1266389), the Leakey Foundation, and the Committee on Research, University of California, Davis to Lynne A. Isbell, and the Wenner-Gren Foundation (grant no. 8386) to Laura R. Bidner; by the Natural Sciences and Engineering Research Council of Canada (NSERC), NAB. PGSD3-404001-2011; by the National Institutes of Health (NIH), WMG. GM83863, and the University of KwaZulu-Natal.

## Author contributions

Niels Blaum & Jonas Stiegler developed the idea; Marlee Tucker & Francesca Cagnacci facilitated data collection, Robert Hering helped analyze the data; Thomas Müller, Marlee Tucker, Niels Blaum, and Jonas Stiegler conducted initial manuscript structuring, internal revision, and outline; Cara A. Gallagher designed the conceptual figure and strongly contributed to revising the manuscript; Nancy Barker, Anne Berger, Niels Blaum, Francesca Cagnacci, Meaghan N. Evans, Cara A. Gallagher, Morgan Hauptfleisch, Robert Hering, Robert Hering, Marco Heurich, Lynne A. Isbell, Stephanie Kramer-Schadt, Thomas Müller, Krzysztof Schmidt, Nuria Selva, Laurel E.K. Serieys, Agnieszka Sergiel, Marlee Tucker, Bettina Wachter, Stephen Webb, Christopher C. Wilmers, Tomasz Zwijacz-Kozica commented on the manuscript. The main persons responsible for providing the data are Nancy A. Barker (*C. crocuta, P. leo*), Floris M. van Beest (*O. moschatus*), Jerrold L. Belant (*C. latrans, C. lupus, U. americanus*), Anne Berger (*C. capreolus, E. europaeus*), Stephen Blake (*B. bison*), Niels Blaum (*A. marsupialis, L. europaeus, T. oryx, T. strepsiceros*), Francesca Brivio (*S. scrofa*), Bayarbaatar Buuveibaatar (*G. subgutturosa*), Francesca Cagnacci (*C. capreolus, C. elaphus, C. ibex, L. lynx, S. scrofa*), Meaghan N. Evans (*V. tangalunga*), Adam W. Ferguson (*G. genetta, I. albicauda*), Claudia Fichtel (*C. ferox, E. rufifrons, P. verreauxi*), Adam T. Ford (*M. guentheri*), Wayne M. Getz (*C. crocuta, P. leo*), Stefano Grignolio (*C. ibex*), Morgan Hauptfleisch (*A. marsupialis, T. oryx, T. strepsiceros*), Robert Hering (*A. marsupialis, T. oryx, T. strepsiceros*), Marco Heurich (*C. elaphus*), Lynne A. Isbell (*C. pygerythrus, P. pardus, P.anubis*), René Janssen (*F. silvestris*), Petra Kaczensky (*E. hemionus*), Sophia Kimmig (*V. vulpes*), Rafał Kowalczyk (*B. bonasus*), Stephanie Kramer-Schadt (*P. lotor, V. vulpes*), Mia-Lana Lührs (*C. ferox*), Jörg Melzheimer (*A. jubatus*), Nicolas Morellet (*C. capreolus*), Manuel Roeleke (*P. lotor*), Christer M. Rolandsen (*A. alces*), Sonia Saïd (*C. capreolus*), Niels M. Schmidt (*O. moschatus*), Nuria Selva (*U. arctos*), Laurel E. K. Serieys (*L. rufus*), Rob Slotow (*P. leo*), Jonas Stiegler (*L. europaeus*), Garrett M. Street (*O. virginianus, S. scrofa*), Wiebke Ullmann (*L. europaeus*), Abi T. Vanak (*C. aureus, F. chaus, V. bengalensis*), Bettina Wachter (*A. jubatus*), Stephen L. Webb (*O. virginianus, S. scrofa*), Christopher C. Wilmers (*P. concolor*). All other co-authors contributed to data collection and approved the final manuscript.

## Funding

## Competing interests

The authors declare no competing interests.

## Additional information

Jonas Stiegler [1,2] ✉, Cara A. Gallagher[1], Robert Hering [1,3], Thomas Müller [4,5,6], Marlee Tucker [7], Marco Apollonio[8], Janosch Arnold[9], Nancy A. Barker[10], Leon Barthel[11], Bruno Bassano[12], Floris M. van Beest[13], Jerrold L. Belant[14], Anne Berger[11], Dean E. Beyer Jr[14], Laura R. Bidner[15,16], Stephen Blake[17,18], Konstantin Börner[11], Francesca Brivio [8], Rudy Brogi [8], Bayarbaatar Buuveibaatar [19], Francesca Cagnacci [20,21], Jasja Dekker [22], Jane Dentinger[23], Martin Duľa [24], Jarred F. Duquette[14], Jana A. Eccard[2], Meaghan N. Evans[25,26], Adam W. Ferguson[16,27], Claudia Fichtel [28], Adam T. Ford[29], Nicholas L. Fowler[14], Benedikt Gehr[30], Wayne M. Getz[31,32], Jacob R. Goheen[33], Benoit Goossens[25,26], Stefano Grignolio [34], Lars Haugaard[13], Morgan Hauptfleisch[35], Morten Heim [36], Marco Heurich [37,38,39], Mark A. J. Hewison[40], Lynne A. Isbell[15,41], René Janssen[22], Anders Jarnemo [42], Florian Jeltsch[1], Jezek Miloš [43], Petra Kaczensky [36,44], Tomasz Kamiński[45], Peter Kappeler [28,46], Katharina Kasper [45], Todd M. Kautz[14], Sophia Kimmig [11], Petter Kjellander[47], Rafał Kowalczyk [45], Stephanie Kramer-Schadt [11,48], Max Kröschel[38], Anette Krop-Benesch [11], Peter Linderoth[9], Christoph Lobas[1], Peter Lokeny[27], Mia-Lana Lührs[28,49], Stephanie S. Matsushima[50], Molly M. McDonough[27], Jörg Melzheimer [11], Nicolas Morellet [40], Dedan K. Ngatia[16], Leopold Obermair[51,52,53], Kirk A. Olson[36], Kidan C. Patanant[54], John C. Payne[19], Tyler R. Petroelje[14], Manuel Pina[55], Josep Piqué[55], Joseph Premier [37,38], Jan Pufelski[1], Lennart Pyritz[28], Maurizio Ramanzin[56], Manuel Roeleke [1], Christer M. Rolandsen [36], Sonia Saïd[57], Robin Sandfort [51], Krzysztof Schmidt[45], Niels M. Schmidt [13,58], Carolin Scholz[1,11], Nadine Schubert [59], Nuria Selva [60,61,62], Agnieszka Sergiel [60], Laurel E. K. Serieys [63], Václav Silovský [43], Rob Slotow [64,65], Leif Sönnichsen[11,45], Erling J. Solberg[36], Mikkel Stelvig[66], Garrett M. Street [67], Peter Sunde[13], Nathan J. Svoboda[68], Maria Thaker [69], Maxi Tomowski [1,70], Wiebke Ullmann[1], Abi T. Vanak [10,71,72], Bettina Wachter[11], Stephen L. Webb [23], Christopher C. Wilmers[50], Filip Zieba[73], Tomasz Zwijacz-Kozica[73] & Niels Blaum [1]

¹Plant Ecology and Nature Conservation, Institute of Biochemistry and Biology, University of Potsdam, 14469 Potsdam, Germany. ²Animal Ecology, Institute of Biochemistry and Biology, University of Potsdam, 14469 Potsdam, Germany. ³Ecology and Macroecology Laboratory, Institute for Biochemistry and Biology, University of Potsdam, 14469 Potsdam, Germany. ⁴Senckenberg Biodiversity and Climate Research Centre, Senckenberg Gesellschaft für Naturforschung, 60325 Frankfurt (Main), Germany. ⁵Department of Biological Sciences, Goethe University, 60438 Frankfurt (Main), Germany. ⁶Smithsonian Conservation Biology Institute, National Zoological Park, Front Royal, VA, USA. ⁷Department of Environmental Science, Radboud Institute for Biological and Environmental Sciences, Radboud University, P.O. Box 9010, 6500 GL Nijmegen, Netherlands. ⁸Department of Veterinary Medicine, University of Sassari, Via Vienna 2, 07100 Sassari, Italy. ⁹Wildlife Research Unit, Agricultural Centre Baden-Wuerttemberg (LAZBW), 88326 Aulendorf, Germany. ¹⁰School of Life Sciences, University of KwaZulu-Natal Durban, South Africa. ¹¹Leibniz Institute for Zoo and Wildlife Research (IZW), Berlin, Germany. ¹²Gran Paradiso National Park, Turin, Italy. ¹³Department of Ecoscience, Aarhus University, Roskilde, Denmark. ¹⁴Department of Fisheries and Wildlife, Michigan State University, East Lansing, MI, USA. ¹⁵Department of Anthropology, University of California, Davis, CA 95616, USA. ¹⁶Mpala Research Centre, 555-10400 Nanyuki, Kenya. ¹⁷Department of Biology, St. Louis University, St. Louis, MO, USA. ¹⁸WildCare Institute, Saint Louis Zoo, 1 Government Drive, Saint Louis, MO 63110, USA. ¹⁹Wildlife Conservation Society, Mongolia Program, Ulaanbaatar, Mongolia. ²⁰Research and Innovation Centre, Animal Ecology Unit, Fondazione Edmund Mach, San Michele all'Adige, Trento, Italy. ²¹NBFC, National Biodiversity Future Centre, Palermo 90133, Italy. ²²Bionet Natuuronderzoek, Stein, Netherlands. ²³Texas A&M Natural Resources Institute, and Department of Rangeland, Wildlife and Fisheries Management, Texas A&M University, College Station, TX 77843-2138, USA. ²⁴Department of Forest Ecology, Faculty of Forestry and Wood Technology, Mendel University, 613 00 Brno, Czech Republic. ²⁵Danau Girang Field Centre, Sabah Wildlife Department, 88100 Kota Kinabalu, Sabah, Malaysia. ²⁶Organisms and Environment Division, School of Biosciences, Cardiff University, Cardiff CF10 3AX, UK. ²⁷Department of Biological Sciences, Chicago State University, 9501 S. King Drive, Chicago, IL 60628, USA. ²⁸German Primate Center, Behavioral Ecology and Sociobiology Unit, 37077 Göttingen, Germany. ²⁹Department of Biology, University of British Columbia, 1177 Research Road, Kelowna, British Columbia, Canada. ³⁰Department of Evolutionary Biology and Environmental Studies, University of Zurich, 8057 Zurich, Switzerland. ³¹Department of Environmental Science Policy & Management, 130 Mulford Hall, University of California at Berkeley, Berkeley, CA 94720-3112, USA. ³²School of Mathematical Sciences, University of KwaZulu-Natal, Private Bag X54001, Durban 4000, South Africa. ³³Department of Zoology and Physiology, University of Wyoming,

Laramie, WY 82071, USA. [34]Department of Life Science and Biotechnology, University of Ferrara, Via Borsari 46, I-44121 Ferrara, Italy. [35]Biodiversity Research Centre, Agriculture and Natural Resources Sciences, Namibia University of Science and Technology, Windhoek, Namibia. [36]Norwegian Institute for Nature Research, P.O. Box 5685 Torgarden, NO-7485 Trondheim, Norway. [37]Department of National Park Monitoring and Animal Management, Bavarian Forest National Park, Freyunger Str. 2, 94481 Grafenau, Germany. [38]Chair of Wildlife Ecology and Management, Faculty of Environment and Natural Resources, University of Freiburg, Tennenbacher Straße 4, 79106 Freiburg, Germany. [39]Institute of Forestry and Wildlife Management, Inland Norway University of Applied Science, NO-2480 Koppang, Norway. [40]Université de Toulouse, INRAE, CEFS Castanet-Tolosan, France. [41]Animal Behavior Graduate Group, University of California, Davis, CA 95616, USA. [42]School of Business, Innovation and Sustainability, Halmstad University, Halmstad, Sweden. [43]Department of Game Management and Wildlife Biology, Faculty of Forestry and Wood Sciences, Czech University of Life Sciences, Kamýcká 129, Prague 6-Suchdol 165 00, Czech Republic. [44]Research Institute of Wildlife Ecology, University of Veterinary Medicine Vienna, A-1160 Vienna, Austria. [45]Mammal Research Institute, Polish Academy of Sciences, Stoczek 1, 17-230 Białowieża, Poland. [46]Department of Sociobiology/Anthropology, University of Göttingen, 37077 Göttingen, Germany. [47]Grimsö Wildlife Research Station, Department of Ecology, Swedish University of Agricultural Sciences, 730 91 Riddarhyttan, Sweden. [48]Institute of Ecology, Chair of Planning-Related Animal Ecology, Technische Universität Berlin, Potsdam, Germany. [49]Büro Renala, Gülper Hauptstr. 4, 14715 Havelaue, Germany. [50]Center for Integrated Spatial Research, Environmental Studies Department, University of California, Santa Cruz, CA 95060, USA. [51]Department of Integrative Biology and Biodiversity Research, University of Natural Resources and Life Sciences, Vienna, Gregor-Mendel-Straße 33, 1180 Vienna, Austria. [52]Department of Integrative Biology and Evolution, Research Institute of Wildlife Ecology, University of Veterinary Medicine, Savoyenstraße 1, 1160 Vienna, Austria. [53]Hunting Association of Lower Austria, Wickenburggasse 3, 1080 Vienna, Austria. [54]Technische Universität München, Arcisstraße 21, 80333 München, Germany. [55]Tragsatec, C. de Julián Camarillo, 6B, San Blas-Canillejas, 28037 Madrid, Spain. [56]Dipertimento di agronomia, animali, alimenti, risorse naturali e ambiente, Università degli Studi di Padova, 35020 Legnaro PD, Italy. [57]Office Français de la Biodiversité, Montfort, 01330 Birieux, France. [58]Arctic Research Centre, Aarhus University, Aarhus, Denmark. [59]Department of Behavioural Ecology, Bielefeld University, Bielefeld, Germany. [60]Institute of Nature Conservation, Polish Academy of Sciences, 31-120 Kraków, Poland. [61]Departamento de Ciencias Integradas, Facultad de Ciencias Experimentales, Centro de Estudios Avanzados en Física, Matemáticas y Computación, Universidad de Huelva, Huelva, Spain. [62]Estación Biológica de Doñana, Consejo Superior de Investigaciones Científicas, Sevilla, Spain. [63]Panthera, 8 W 40th St, 18th Floor, New York, NY 10018, USA. [64]Amarula Elephant Research Programme, School of Life Sciences, University of KwaZulu-Natal, Durban 4041, South Africa. [65]Department of Genetics, Evolution and Environment, University College, London WC1E 6BT, UK. [66]Copenhagen Zoo, Frederiksberg, Denmark. [67]Department of Wildlife, Fisheries, and Aquaculture, Mississippi State University, Mississippi State, MS, USA. [68]Alaska Department of Fish and Game, Wildlife Division, 11255 W. 8th Street, AK, USA. [69]Center for Ecological Sciences, Indian Institute of Science, Bengaluru 560012, India. [70]Evolutionary Biology / Systematic Zoology, Institute of Biochemistry and Biology, University of Potsdam, Potsdam, Germany. [71]Centre for Biodiversity and Conservation, Ashoka Trust for Research in Ecology and the Environment, Bangalore, India. [72]Wellcome Trust/DBT India Alliance, Clinical and Public Health Program, Bengaluru, India. [73]Tatra National Park, Zakopane, Poland.
✉e-mail: stiegler@uni-potsdam.de

