## [Peer Review File · Nature Communications]

Mammals show faster recovery from capture and tagging in human-disturbed landscapesREVIEWER COMMENTS

Reviewer #1 (Remarks to the Author):

Review of "Sensitivities of mammals to capture and tagging...".

This is a nice analysis of tracking data around an important question by an impressive group of authors. Overall, I think this is well analyzed and written, I have a few minor writing suggestions below.

My only gripe is that they didn't do more with these data.

For example, the effect of method of capture, type of tag, and drug combinations are not evaluated. It sounds like they were aware of these ideas but didn't have enough data, which is surprising, but I guess there's nothing we can do about it now. A bit more explanation would still be useful.

There are two other analyses that immediately come to mind that they could add. First – I would like some mention of mortality – did any animals die in the first 10-20 days? More/less than expected? I didn't see any mention of this at all. Second, it would be super interesting to see a movement analysis (e.g. SSF) to check if animals avoid their site of capture, since this could have big impacts on subsequent results, and could last longer than 4-7 days. Given the big author list, with lots of expertise, this might not be asking too much.

Abstract – generally good, there are a few passive sentences I'd rather see rewritten in active voice but that's just my preference.

Introduction

-L31 – I don't think the paragraph describing ACC is needed.

-L68 – this avoidance of activity center brings up a new question of rather animals avoid

their capture site. It would be great if you could add that analysis to your study, should be straight forward with SSF.

-L92 – I don't understand the assumption that larger animals get stronger anesthetic. Dosage is always per kg of the animal. This needs more explanation and/or a citation.

L92-> This last paragraph gets a bit long, but I think actually needs to be longer. I would split it about here and point out that the activity of tagging is actually a manipulative experiment in itself that allows us to ask a set of questions about how animals respond to this kind of disturbance. So past the methodological question of what data to throw out, we can ask some biology questions. Then lay out the questions but expand a bit and include some citations to justify your hypotheses.

Table 2 – not clear what SD means in this table, is this standard deviation? Better to add that to the mean after +/-

Figure 3 – overall a nice graphs but I'd like to see units on the Y axis so we can evaluate the biological significance of these differences. I'm also not sure A and B are both needed since they show the same information, but it doesn't take up a lot more room, so it probably doesn't matter.

Discussion

- L150 please also point out how many species this assessment is new for.

- L156 – state how many days faster on average

- L174 - as in the intro, more explanation on these chemical differences please. Are they different classes of drugs?

- L198 – I think you need to mention in the discussion why you did not evaluate deployment type, since the methods in this paper are stuck at the bottom. I also think you can revise this text a bit to suggest it could be an interactive effect with HFI, and so your HFI results are tentative.

- L203 – I'm not really sure of the purpose of this paragraph. Consider revising or removing.

- Conclusions. I think you can be a bit firmer on two things. First, to me, these impacts seem

really small and short lived, which suggests collaring overall has a very low impact on most animals, and I think you can come out and say that stronger. This is good news for animal tracking science. Second, any short-term (<7d) studies are going to be severely compromised. I think you can come out harder on this, but also provide some examples – what studies have published short term data? What kind of research requires this? Please provide more details. Maybe very small animals? Or manipulative experiments (eg. navigation)?

Reviewer #2 (Remarks to the Author):

Thank you for this contribution. We must be aware of how handling animals influences their behaviour and survival afterwards. The public demands that we be conscientious. This broad-based synthesis is crucial for our continued research on wild animals. The article is well written, conclusions are sound and I support publication.

Reply to the Reviewers

Re: Manuscript # NCOMMS-24-04377

“Sensitivities of mammals to capture and tagging: faster recovery in human-disturbed landscapes”

Stiegler et al.

Dear Reviewers,

We thank you for the opportunity to submit a revised version of our manuscript, entitled “Sensitivities of mammals to capture and tagging: faster recovery in human-disturbed landscapes”, and for the effort you put into enhancing our work. Thanks to your thorough and helpful comments, we have made substantial improvements to our manuscript. Please find our point-by-point response to your comments and suggestions below. Further, please find attached a document with tracked changes, to which the line numbers given here refer to.

Reviewer #1, comment #1

This is a nice analysis of tracking data around an important question by an impressive group of authors. Overall, I think this is well analyzed and written, I have a few minor writing suggestions below. My only gripe is that they didn't do more with these data. For example, the effect of method of capture, type of tag, and drug combinations are not evaluated. It sounds like they were aware of these ideas but didn't have enough data, which is surprising, but I guess there's nothing we can do about it now. A bit more explanation would still be useful.

Our response #1.1

Many thanks for your time, the thorough reading, and the positive response to our manuscript. Indeed, we originally hoped to additionally investigate the effects of the capture method, the tag itself, and the impact of the different sedatives. Regrettably, the documentation of these details fell short of our expectations. Despite our efforts to collect as much data as possible, we still ended up with comparatively small datasets in cases where our response variable had numerous levels. For example, from the drug compositions, we could see that almost every veterinarian had more or less their own unique field-approved mix, making comparison

difficult. Tag type was mostly documented, yet information on its specific weight and antenna placement - whether protruding outside or hidden within the collar - could no longer be traced. Additionally, since individuals were often not weighed, we could not properly calculate a collar-to-body mass ratio. Considering this, we focused on the most reliable data to generate meaningful results.

Reviewer #1, comment #2

There are two other analyses that immediately come to mind that they could add. First, I would like some mention of mortality – did any animals die in the first 10–20 days? More/less than expected? I didn't see any mention of this at all. Second, it would be super interesting to see a movement analysis (e.g. SSF) to check if animals avoid their site of capture, since this could have big impacts on subsequent results, and could last longer than 4–7 days. Given the big author list, with lots of expertise, this might not be asking too much.

Our response #1.2

We thank you for these suggestions and agree that reporting mortality or quantifying mortality rates after handling and collaring would be valuable. Since we were focusing on the sensitivities of mammals in activity and movements, we unfortunately only collected data from individuals that survived the entire 20-day period without any data gaps or missing information. Therefore, all individuals in this study survived for the entire study duration, and we do not have information on additional individuals that may have died within the first 20 days. Although appealing, reporting mortality data related to collaring and handling is very sensitive information that not all data providers may be willing to share.

Regarding your second point about conducting a step selection function (SSF) to determine if animals avoided their capture site, we acknowledge the potential value of such an analysis in enhancing our understanding of the immediate reactions of animals to capture. However, after careful consideration, we believe this analysis requires additional data that were not collected. Several factors could strongly influence the likelihood of animals revisiting their capture sites. These factors include, but are not limited to, the current migratory or stationary status of individuals, the duration of their home range crossing time if they are stationary, whether females are currently caring for offspring, and the territorial status of individuals (e.g., whether they have established their own territory or are still 'floaters,' as is common in species like cheetahs). Without further information, any conclusions drawn from the SSF analysis could be significantly skewed or incomplete.

Another important factor would be the capture type. For example, in the case of hares, we used two different capture techniques: identifying and capturing hares at their resting spots and setting up traps in areas where hares are actively found. Analyzing the distance of hares from their release location over 20 days, we observed that some individuals returned to the capture site while others did not (Fig. 1, left). The second example illustrates a similar pattern in the case of male lynx (Fig. 1, right). Trap-happy and returning individuals were

also regularly observed in the urban fox project and might reflect personality differences (S KS, personal information).

Figure 1: Distance of individuals to their release location. Left: female hares; right: male lynx

However, hare and lynx individuals that do not return might be dispersing to new habitats, making it difficult to attribute this behavior solely to the collaring event. Given the number of species and more than 1500 individuals from 42 species in our dataset, it would be challenging to account for each study animal's specific life history and behavior. In addition, the level of detail recorded for each individual varies across the different studies. Therefore, we believe that conducting an SSF analysis to assess avoidance behavior would be more appropriate as a separate, focused study because the whole movement trajectory would be needed for drawing conclusions on the differences in space use compared to the first 20 days. Our current study aims to provide reliable data with sound implications, and adding this additional layer of analysis might detract from that focus.

We appreciate the suggestion, though, and acknowledge its potential to contribute valuable insights in a dedicated follow-up study.

Reviewer #1, comment #3

Abstract – generally good, there are a few passive sentences I'd rather see rewritten in active voice but that's just my preference.

Our response #1.3

Many thanks for this; we rewrote the Abstract in active voice.

Reviewer #1, comment #4

L31 – I don't think the paragraph describing ACC is needed.

Our response #1.4

We agree that the paragraph was too long/detailed, yet we also feel that we need to sufficiently introduce accelerometers and ODBA as they are central to our study and the respective analysis. Hence, we shortened this paragraph and merged it with to the previous one; it now reads: "[.] accelerometers. Such sensors measure static and dynamic acceleration, representing animal movement in three dimensions [33] and allowing for the quantification of activity or proxies thereof [78, 57, 41, 27]. One well-established proxy is the overall dynamic body acceleration (ODBA), which estimates activity-related energy expenditure of free-ranging animals [81, 24]." lines 32-36.

Reviewer #1, comment #5

L68 – this avoidance of activity center brings up a new question of rather animals avoid their capture site. It would be great if you could add that analysis to your study, should be straight forward with SSF.

Our response #1.5

As stated previously, we very much like your suggestion of an additional SSF analysis, but we do not see it as feasible in this manuscript. Please see our more detailed answer to your comment # 2 response #1.2

Reviewer #1, comment #6

L92 - I don't understand the assumption that larger animals get stronger anesthetic. Dosage is always per kg of the animal. This needs more explanation and/or a citation.

Our response #1.6

Many thanks for raising this point; you are correct. However, although anesthetic dosage is adjusted per kilogram, larger mammals often require a longer duration of anesthesia and face more significant physiological challenges during immobilization. This can lead to heightened stress and more prolonged recovery times than in smaller mammals; for instance, larger animals have more complex thermoregulatory challenges, making them more susceptible to hypothermia during and after anesthesia. The need for remote drug delivery methods (e.g., darting), which are more commonly used for larger animals, can also introduce additional stress and potential for injury, further complicating recovery (Chinnadurai et al. 2016). Furthermore, larger mammals typically exhibit more pronounced behavioral changes post-anesthesia due to the extended duration of immobilization and the higher risk of physiological complications. Studies have shown that an animal's size, habitat, and activity level significantly influence the anesthetic protocol and its impact, with larger species facing longer recovery periods (Powell & Proulx, 2003).

Sathya K. Chinnadurai, Danielle Strahl-Heldreth, Christine V. Fiorello, Craig A. Harms; Best-Practice Guidelines for Field-Based Surgery and Anesthesia of Free-Ranging Wildlife. I. Anesthesia and Analgesia. *J Wildl Dis* 1 April 2016; 52 (2s): S14–S27. doi: <https://doi.org/10.7589/52.2S.S14>

Roger A. Powell, Gilbert Proulx, Trapping and Marking Terrestrial Mammals for Research: Integrating Ethics, Performance Criteria, Techniques, and Common Sense, *ILAR Journal*, Volume 44, Issue 4, 2003, Pages 259–276, doi: <https://doi.org/10.1093/ilar.44.4.259>

To further clarify this point, we have revised the respective passage (lines 92-98) as: "Even though anesthesia dosage is calculated per kilogram, we assumed that larger mammals experience a more pronounced disturbance, as they often require longer durations of anesthesia [55] and face more significant physiological challenges during immobilization such as hyperthermia [16] compared to smaller mammals. [..]"

Reviewer #1, comment #7

L92 - This last paragraph gets a bit long, but I think actually needs to be longer. I would split it about here and point out that the activity of tagging is actually a manipulative experiment in itself that allows us to ask a set of questions about how animals respond to this kind of disturbance. So past the methodological question of what data to throw out, we can ask some biology questions. Then lay out the questions but expand a bit and include some citations to justify your hypotheses.

Our response #1.7

We appreciate your suggestion to split and expand the final paragraph of the introduction, emphasizing that the activity of tagging is a manipulative experiment in itself. Before starting with our hypotheses, the revised section of this paragraph now starts with (lines 99-101):

”Capturing and tagging animals acts as a manipulative experiment, enabling us to investigate how animals respond to such disturbances. Beyond data exclusion considerations, such studies allow for the exploration of various hypotheses related to patterns in the behavioral responses of animals.”

Reviewer #1, comment #8

Table 2 – not clear what SD means in this table, is this standard deviation? Better to add that to the mean after +/-

Our response #1.8

Yes, SD means standard deviation here; we agree and modify the table according to your suggestion. Please see the revised table 2 at line 153.

Reviewer #1, comment #9

Figure 3 – overall a nice graphs but I'd like to see units on the Y axis so we can evaluate the biological significance of these differences. I'm also not sure A and B are both needed since they show the same information, but it doesn't take up much more room, so it probably doesn't matter.

Our response #1.9

Many thanks for your comment. Regarding the presentation of units, we opted to remove the axis ticks and keep it as a slow-fast continuum for clarity, as the unit of recovery speed is rather uncommon: the unit of the calculated slope is ”Adaption to long-term mean [%/day]”. This reflects how quickly the percent’s activity/displacement is changing after the release, i.e., this unit measures the speed of change as a percentage of the long-term mean activity on day 1 and quantifies the rate of adaptation of individuals. To make this clearer, we revised the respective description in the methods section (lines 290-291) and specified the Y axis in Figure 3 accordingly. Following your suggestion, the revised figure has improved considerably.

Concerning the necessity of Fig. 3B, we feel that this helps the reader to understand both a) the differing landscapes of areas with different HFI values and b) the sex-specific recovery speed more directly. However, we also agree that this specific subplot is not essential to follow the overall manuscript and are willing to remove it if the editor also suggests doing so.

Reviewer #1, comment #10

L150 please also point out how many species this assessment is new for.

Our response #1.10

We added the requested information to the discussion (lines 164-165): "To our knowledge, this study is the first to characterize post-capture behavior in the other mammalian species analyzed."

Reviewer #1, comment #11

L156 – state how many days faster on average

Our response #1.11

Many thanks; we completely agree that this point needs to be stated. Accordingly, we modified the respective paragraph. It now starts with "Males recovered on average 1.3 days faster than females from collaring-induced changes in activity, [..]", line 166

Reviewer #1, comment #12

L174 - as in the intro, more explanation on these chemical differences please. Are they different classes of drugs?

Our response #1.12

Please see our response to your previous comment # 6 response #1.6 and the according changes in the introduction.

Reviewer #1, comment #13

L198 – I think you need to mention in the discussion why you did not evaluate deployment type, since the methods in this paper are stuck at the bottom. I also think you can revise this text a bit to suggest it could be an interactive effect with HFI, and so your HFI results are tentative.

Our response #1.13

We followed your suggestion and changed the end of the paragraph as follows (lines 209-212): "With numerous deployment methods like helicopter darting, chasing or trapping being applied in the field, analyzing their effect was not feasible within the scope of this study. The effect of the deployment method remains unclear, and the selected method may even change along an HFI gradient." And further (lines 213-215): "Interpreting the effect of the human footprint should take into account that the deployment method could be influenced by the respective study area. Therefore, we strongly recommend documenting these methodological decisions for future research."

Reviewer #1, comment #14

L203 – I'm not really sure of the purpose of this paragraph. Consider revising or removing.

Our response #1.14

Thanks for your suggestion. After careful revision, we decided to remove two sentences from this paragraph. We feel that this section is crucial because it underscores the ethical considerations necessary in wildlife tracking, emphasizing the need to balance data collection with animal welfare. Additionally, it highlights the potential long-term impacts of short-term behavioral changes, particularly in terms of energetics, which are essential for accurately informing conservation strategies. We also backed up our statements with relevant literature. Nevertheless, we are happy to remove it if the editor wishes.

Reviewer #1, comment #15

Conclusions. I think you can be a bit firmer on two things. First, to me, these impacts seem really small and short lived, which suggests collaring overall has a very low impact on most animals, and I think you can come out and say that stronger. This is good news for animal tracking science. Second, any short-term (<7d) studies are going to be severely compromised. I think you can come out harder on this, but also provide some examples – what studies have published short term data? What kind of research requires this? Please provide more details. Maybe very small animals? Or manipulative experiments (eg. navigation)?

Our response #1.15

Thank you for motivating us to be more bold in our conclusions. We revised our conclusions to incorporate your suggestions. Specifically, we added, e.g., "These impacts, however, fade within a relatively short time frame of four to seven days, suggesting that the overall impact of collaring is minimal and short-lived, which is good news for animal tracking science." (lines 232-235); and lines (235-238): In studies where longer tracking is not feasible, researchers should be aware of these disturbance biases. Particularly, short-term studies, lasting less

than seven days, may be significantly compromised. These studies are prevalent in certain research areas, for example, where battery weight strongly limits tracking duration.”

Reviewer #2, comment #1

Thank you for this contribution. We must be aware of how handling animals influences their behaviour and survival afterwards. The public demands that we be conscientious. This broad-based synthesis is crucial for our continued research on wild animals. The article is well written, conclusions are sound and I support publication.

Our response #2.1

Thank you for your supportive and thoughtful feedback. We appreciate your acknowledgment of the importance of understanding how handling and collaring can impact animal behavior. We are glad that you found our synthesis broad-based and our conclusions sound.

Postscript

We would like to thank the two anonymous reviewers for the constructive recommendations and suggestions that helped us to improve our manuscript substantially!